# Functional Screening Techniques to Identify Long Non-Coding RNAs as Therapeutic Targets in Cancer

**DOI:** 10.3390/cancers12123695

**Published:** 2020-12-09

**Authors:** Kathleen M. Lucere, Megan M.R. O’Malley, Sarah D. Diermeier

**Affiliations:** Department of Biochemistry, University of Otago, Dunedin 9016, New Zealand; lucka387@student.otago.ac.nz (K.M.L.); omame774@student.otago.ac.nz (M.M.R.O.)

**Keywords:** long non-coding RNA (lncRNA), cancer, oncology, CRISPR, functional screening, siRNA, shRNA, antisense oligonucleotide (ASO), target identification, therapeutic target

## Abstract

**Simple Summary:**

Long non-coding RNAs (lncRNAs) are a recently discovered class of molecules in the cell, with potential to be utilized as therapeutic targets in cancer. A number of lncRNAs have been described to play important roles in tumor progression and drive molecular processes involved in cell proliferation, apoptosis or invasion. However, the vast majority of lncRNAs have not been studied in the context of cancer thus far. With the advent of CRISPR/Cas genome editing, high-throughput functional screening approaches to identify lncRNAs that impact cancer growth are becoming more accessible. Here, we review currently available methods to study hundreds to thousands of lncRNAs in parallel to elucidate their role in tumorigenesis and cancer progression.

**Abstract:**

Recent technological advancements such as CRISPR/Cas-based systems enable multiplexed, high-throughput screening for new therapeutic targets in cancer. While numerous functional screens have been performed on protein-coding genes to date, long non-coding RNAs (lncRNAs) represent an emerging class of potential oncogenes and tumor suppressors, with only a handful of large-scale screens performed thus far. Here, we review in detail currently available screening approaches to identify new lncRNA drivers of tumorigenesis and tumor progression. We discuss the various approaches of genomic and transcriptional targeting using CRISPR/Cas9, as well as methods to post-transcriptionally target lncRNAs via RNA interference (RNAi), antisense oligonucleotides (ASOs) and CRISPR/Cas13. We discuss potential advantages, caveats and future applications of each method to provide an overview and guide on investigating lncRNAs as new therapeutic targets in cancer.

## 1. Introduction

In recent years, The Cancer Genome Atlas (TCGA) and other large-scale cancer genomics projects have revealed that many cancer-associated mutations are located in non-coding regions of the genome, including non-coding RNA genes [1,2,3,4,5]. Long non-coding RNAs (lncRNAs) comprise the largest and most diverse class of non-coding transcripts, with over 100,000 transcripts identified to date [6]. They are defined by length (>200 nt) and can be capped, spliced and polyadenylated but have limited protein-coding potential [7,8]. Many lncRNAs are expressed in a distinct spatial, temporal, tissue- and cell type-specific manner, including tumor specific expression [9,10,11]. In addition, recent studies revealed that numerous lncRNAs act as key drivers of cell proliferation and/or invasive potential, making them promising new targets for systemic cancer therapy (as reviewed in [12,13]).

Originally, cancer associated lncRNAs were often identified based on differential expression between tumors and corresponding normal tissue. However, differential expression in itself does not imply that a lncRNA is a genuine driver or suppressor of tumor cell growth [11]. Functional analysis of candidate lncRNAs is necessary for identifying viable therapeutic targets. While individual characterization has successfully revealed a number of lncRNAs for therapeutic targeting (as reviewed in [13]), recent technical advances allow for high-throughput screens to test the functionality of thousands of lncRNAs in parallel.

Here, we summarize state-of-the-art functional screening methods capable of altering genomic, transcriptomic, or post-transcriptional regulation. Functional screens can generally be categorized into gain-of-function (GOF) and loss-of-function (LOF) models [14,15]. Screen readout is accomplished by measuring a predetermined phenotype, such as cellular viability or resistance to drug treatments [14,16]. Libraries targeting select lncRNAs are often delivered as a pool while ensuring that only one lncRNA is modified in each cell, for example by using a comparably low multiplicity of infection (MOI) for viral delivery of the library [17]. Upon completion of a screen, enrichment or depletion of targets is assessed via high-throughput sequencing of the corresponding LOF or GOF constructs integrated into the genome of the screening model. Analysis of negative or “drop-out” screens seek LOF constructs that are decreased, indicating the targeted gene plays a role in the intended phenotype, such as cancer viability or sensitivity to drug treatment (as reviewed in [18]). Positive screens aim to identify LOF constructs that remain or increase throughout the screen, also indicating the role of the targeted gene in the observed phenotype, such as conferring resistance in the presence of cancer therapeutics (as reviewed in [18]). The resulting loss or gain of targeted lncRNAs indicate their functional role in cancer. In this review, we highlight the current functional screening techniques available for lncRNAs and their applications in cancer research (Table A1).

## 2. Genomic and Transcriptional Targeting

Recent studies identified numerous CRISPR/Cas systems suitable for diverse applications of genomic and transcriptional regulation. The genomic knockout of a target gene is facilitated through catalytically active Cas proteins, whereas transcriptional regulation is achieved through nuclease deficient Cas9 (dCas9) proteins fused to transcriptional repressor or activator domains [17,19,20]. Target gene expression can be attenuated using CRISPR interference (CRISPRi) or up-regulated through CRISPR activation (CRISPRa) [15,16,17] (Figure 1). Here, we will discuss each of these CRISPR systems and their application for functional screening of lncRNAs in cancer.

### 2.1. CRISPR/Cas9 Knockout

Large-scale CRISPR/Cas9 functional screens have been widely performed to identify protein-coding genes as therapeutic targets in cancer [21,22,23], whereas only two studies applied this technique to lncRNAs thus far [24,25]. For protein-coding genes, Cas9 is usually directed by a target-specific single guide RNA (sgRNA) to the open reading frame (ORF) of a gene of interest, resulting in a blunt double-stranded break (DSB) in the DNA [20]. This triggers the non-homologous end joining repair pathway, which introduces premature stop codons through frameshift indel mutations, effectively disrupting the ORF of the target messenger RNA (mRNA) [20]. In most cases, disruption of the ORF results in either degradation of the mutated transcript through nonsense-mediated mRNA decay, or translation of non-functional truncated proteins [19,26].

This principle is not directly applicable to lncRNAs as these usually contain no or short, non-functional ORFs, which creates challenges for sgRNA design [27,28]. Unlike protein-coding genes in which disruption of the ORF is well known to affect function, it is currently not known for most lncRNAs which parts of the sequence are functionally important. Additionally, currently available lncRNA annotations are incomplete and often inaccurate compared to protein-coding genes, in particular with regards to the correct annotation of the transcription start site (TSS), creating challenges for the genomic targeting of lncRNAs [29]. As lncRNA function is often determined by the secondary structure of the transcript, small deletions within lncRNA genes may not always lead to a complete knockout [30]. Due to the current gap of knowledge, it is unclear if and how lncRNA expression can be sufficiently disrupted by the catalytically active CRISPR/Cas9 complex.

#### 2.1.1. CRISPR/Cas9 Paired-Guide RNA Approach

The above discussed limitations of catalytically active Cas9 for functional screens of lncRNAs can be overcome using a paired-guide RNA (pgRNA) approach. LncRNA genes can be successfully knocked out using a pair of sgRNAs designed to bind upstream and downstream of the promoter/TSS, leading to the deletion of comparably long stretches of DNA [24,30,31,32,33] (Figure 1A). The pgRNAs can also be utilized to delete an entire locus [30,32,33].

In 2016, Zhu et al. published the first large-scale functional screen of human lncRNAs in cancer using a pgRNA library targeting either the protomer, the promoter and the first exon, or the gene body of the target lncRNAs [24]. They used a dual gRNA lentiviral vector backbone plasmid to ensure delivery of the pgRNAs into the cell at the same time [24]. The pgRNA library was designed to target 671 human lncRNA genes that had previously been implicated in cancer or other diseases, with up to 20 pgRNAs per gene, in three biological replicates [24]. In addition, 17 ribosomal genes and 3 cancer-related genes (*FOXA1*, *HOXB13* and *EZH2*) were targeted as positive controls (100 pgRNAs per gene) [24]. Three types of negative controls were used: non-targeting pgRNAs (100 pairs), pgRNAs targeting the non-functional adeno-associated virus integration site 1 (*AAVS1*; 100 pairs) and pgRNAs targeting the introns of positive control genes (300 pairs) [24]. Following 30 days of growth in the liver cancer cell line Huh7.5_OC_, end point samples were compared to those at day of infection to identify lncRNAs that either suppress or promote cell growth [24]. The study identified 43 lncRNAs that negatively affect cell viability, and eight that promote cell growth [24]. The authors further validated nine of their top hits that did not overlap protein-coding genes using CRISPRi or CRISPRa, as well as by generating individual CRISPR/Cas9 knockouts using the pgRNA approach [24].

CRISPRi and CRISPRa are valuable strategies for validation because they utilize dCas9 proteins, which control for any effect that active Cas9 may have on the observed phenotype, such as DNA repair mechanisms [34,35]. One of the caveats of the pgRNA approach is the relatively variable and difficult to predict deletion efficiency [24], which creates challenges for scaling of the method to potentially thousands of targets. Specifically, the researchers recommended >20 pgRNAs per gene to account for this variability, which is at least twice the number of sgRNAs commonly used in most CRISPR screening approaches [24]. Furthermore, the sensitivity of this screening approach has been questioned by Bergadà-Pijuan et al., who compared the results of Zhu et al. to the Liu et al. CRiNCL screen (as discussed below), which targeted 281 of the same lncRNAs in HeLa cells [36]. Their detailed analysis found the CRISPRi screen to be superior in regards to accuracy and sensitivity in identifying cancer related lncRNA hits compared to the pgRNA library [36].

#### 2.1.2. CRISPR/Cas9 Targeting of Splice Sites

The Wei Laboratory developed a model for Cas9 targeting of splice sites as an alternative approach to the pgRNA method [25]. In this technique, sgRNAs guide Cas9 to the splice sites of genes to efficiently cause exon skipping or intron retention [25] (Figure 1B). The sgRNAs are designed to target either the 5′ splice donor or 3′ splice acceptor site of a lncRNA gene [25] (Figure 1B). They designed a library containing 126,773 sgRNAs targeting splice sites of 10,996 lncRNA genes, along with 500 non-targeting negative control sgRNAs and 350 positive control sgRNAs targeting 36 essential ribosomal genes [25]. This is a substantial increase in library size compared to their previous pgRNA approach (671 lncRNAs targets), indicating the comparably easier scalability of the splice-targeting approach [24,25]. They performed two biological replicates of their screen in the chronic myeloid leukemia (CML) cell line K562 [25]. After 30 days in culture, sgRNA dropout was assessed between the experimental library and non-targeting controls, revealing a total of 230 lncRNAs as essential for K562 cell growth [25]. The top 35 lncRNA hits were chosen for validation using independent splice-targeting models, and 14 lncRNAs were investigated in pgRNA deletion models [25]. All 41 of the validated lncRNAs were confirmed as essential for K562 cell viability [25].

The authors further carried out splice-targeting screens in two additional cell lines, the cervical cancer cell line HeLa and the lymphoblastoid cell line GM12878 [25]. They identified 220 lncRNAs to be involved in cell growth in GM12878 cells and 115 in HeLa cells [25]. Most interestingly, only 20 lncRNAs were identified in all three cell lines, demonstrating the cell type specificity of lncRNA function, as well as the selectivity of this screening approach [25].

It is important to note that 14,470 lncRNAs were initially identified as potential screen targets, however, 2477 were excluded due to the lack of splice sites [25]. This highlights a caveat of the method as not all lncRNAs can be targeted using this approach. Many lncRNAs are monoexonic, including some of the most well characterized ones in cancer, such as *MALAT1* and *NEAT1* [37,38,39]. Additionally, the authors analyzed four of the lncRNA hits that had been validated by both splice-targeting and pgRNA approaches with CRISPRi [25]. Only one of the four lncRNAs was found to be essential for K562 cell viability when targeted with CRISPRi [25]. The authors credit this discrepancy to CRISPRi decreasing gene expression rather than generating a complete gene knockout [25]. Alternatively, the differences in the observed phenotype could be due to Cas9 nuclease activity in the splice-targeting and pgRNA approaches [34,40]. Moreover, it is debatable whether non-targeting negative control sgRNAs are sufficiently controlling for the potentially toxic Cas9 nuclease activity, or if targeting known, non-functional regions of the genome would be more appropriate [41,42]. The discrepancies observed using Cas9 compared to dCas9 approaches highlight the importance of using several different methods to validate the observed phenotype of hits from functional screens.

In addition, both methods utilizing catalytically active Cas9 must account for the effect of nuclease activity on regions with copy number alterations (CNAs). It has been shown that targeting of Cas9 to such regions can result in multiple DSBs, which negatively impacts proliferation in a manner that is independent of target gene function [42,43]. This effect may result in the generation of false positive hits, especially in the case of cancer cells that often harbor CNAs [42,43]. For this reason, it is advisable to filter for regions affected by CNAs when designing catalytically active Cas9 libraries, as well as validation of all downstream hits with appropriate orthogonal approaches such as CRISPRi, which does not generate DSBs, or methods directly targeting transcripts as described in Section 3 below. Details of potential false positives in the Wei Laboratory’s splice-site targeting screen [25] as a result of targeting CNA regions, and overlap with protein-coding genes is reviewed in detail in Horlbeck et al. [40].

Overall, CRISPR/Cas9 screens are a valuable asset for functional analysis of lncRNAs in cancer. Limitations of this method include potential false positives as a result of Cas9 nuclease activity and targeting of regions affected by CNAs, laborious scale-up of the pgRNA approach, and inability to target monoexonic lncRNAs by the splice-targeting method. Alternative approaches such as dCas9 screens are described below.

### 2.2. CRISPRi

CRISPRi is a modified CRISPR system used for transcriptional repression as opposed to genomic editing [17] (Figure 1C). Although dCas9 alone acts as a roadblock to inhibit transcription, repressor domains fused to dCas9 are able to recruit chromatin-modifying complexes and further improve transcriptional repression [44]. The dCas protein is commonly fused to the Krüppel-associated box (KRAB) domain, although other repressor domains such as the SIN3-interacting domain (SID), the WRPW domain of Hes1, and the chromo shadow domain of HP1α have been investigated as well [44,45]. Recently, Alerasool et al. identified the ZIM3-KRAB domain coupled to dCas9 as the most efficient repressor out of 57 KRAB domains assayed in human embryonic kidney 293T (HEK293T) and K562 cell lines [46].

Epigenomic modifications of dCas9-KRAB activity can be measured by H3K9me3 modification to the target locus, thereby reducing chromatin accessibility [14,47]. The activity of CRISPRi can be reversed and is associated with fewer unwanted effects than actively modifying the genomic sequence, such as cleavage induced proliferation arrest [17,41]. CRISPRi is usually achieved by recruiting dCas9-KRAB to the TSS of the target gene. While CRISPRi is able to act within a 1 kb window around the target site, sgRNAs binding within a region of −50 bp upstream to +300 bp downstream of the TSS were found to promote the strongest repression, with maximum activity between +50 bp to +100 bp of the TSS [14,35,45,48]. CRISPRi screens have successfully identified lncRNAs driving cancer cell growth, as well as lncRNAs contributing to the response of the cell to anti-cancer drugs [14,45,48].

#### 2.2.1. CRiNCL Screen for Drivers of Cell Growth

The Weissman and Lim laboratories paved the way for CRISPRi screening by developing the CRISPRi Non-Coding Library (CRiNCL) [14]. They merged non-coding transcriptome annotations from different sources including the Broad human lincRNA catalog, Ensemble build 75, the MiTranscriptome, and previously identified lncRNAs specific to the human brain [14]. Targeted lncRNAs were prioritized based on their expression in RNA-seq data of at least one of several cell lines, including glioblastoma (U87), K562, HeLa, mammary adenocarcinoma (MCF 7 and MDA-MB-231), HEK293T, human foreskin fibroblasts (HFF), and induced pluripotent stem cells (iPSC) [14]. The full CRiNCL library targeted 16,401 lncRNA TSSs with 10 sgRNAs per TSS, for a total library size of 170,262 sgRNAs [14]. This included randomly generated non-targeting sgRNAs weighed by nucleotide frequencies of the targeting sgRNAs, and filtered for no genomic target sites as negative controls [14]. The library was divided into 13 sub-libraries based on overlapping cell line expression, and combinations of each sublibrary including a proportionate amount of non-targeting controls were screened in the aforementioned cell lines excluding HFF cells [14]. Two replicates were performed in each cell line, except for HEK293T cells, in which only one replicate was performed [14].

For the analysis, sgRNAs significantly enriched or depleted as determined by a stringent pipeline including a false discovery rate (FDR) of the non-targeting controls were indicative of lncRNA hits functioning to increase or decrease cell growth, respectively. Out of the 668 lncRNA hits, 169 were removed due to the target TSS positioned within 1 kb of the TSS of a protein coding gene previously deemed essential, resulting in a total of 499 lncRNA hits [14]. Of these, 299 were farther than 1 kb from any protein coding gene, to which the authors state approximately 90% of the protein coding genes proximal to the remaining 200 hits would not affect growth upon knockdown [14]. Interestingly, 89.4% of the total hits and 82.6% of hits expressed in all seven cell lines had a phenotypic effect specific to only one cell line, and not one lncRNA altered cell growth across all seven cell lines [14].

The authors chose to investigate *LINC00263* in more detail, which despite being expressed in all seven cell lines had a more robust effect negatively impacting U87 cell growth [14]. There was no correlation between abundance of *LINC00263* and observed growth effects, suggesting a functional role in U87 cells only [14]. Individual CRISPRi assays validated this observation by directing two distinct sgRNAs targeting *LINC00263* in K562, MCF7, HeLa, and U87 cells, which indeed reduced U87 cell growth only [14]. The authors measured chromatin modifications by chromatin immunoprecipitation sequencing (ChIP-seq), which confirmed equal enrichment levels of H3K9me3 at the *LINC00263* promoter and, therefore, equivalent dCas9-KRAB activity in the cell lines tested [14]. Transcriptomic analysis revealed similar knockdown efficiencies of *LINC00263* in U87, K562, and HeLa cells. However, only U87 cells showed differential expression of additional genes, revealing the up-regulation of ER stress and apoptosis [14]. The authors further validated their CRISPRi findings by knocking-down *LINC00263* with antisense oligonucleotides (ASOs) in U87 and HeLa cells, which again resulted in a decreased growth phenotype in U87 cells only [14]. Based on their results, the authors conclude that irrespective of expression, lncRNA function is cell line specific [14].

#### 2.2.2. CRiNCL Screen for Radiation Sensitizers

In addition to identifying lncRNAs that drive cancer growth, CRISPRi screens can be used to identify vulnerabilities in combination with the standard of care, such as chemotherapy or radiotherapy. The Lim laboratory, the same group that co-developed the CRiNCL library described above, sought to identify lncRNAs as therapeutic targets for sensitization of human glioblastoma (GBM) to radiotherapy [48]. They applied CRiNCL sub-libraries targeting 5689 lncRNAs to U87 cells exposed to a clinically relevant radiation dose (8 Gy delivered in 4 fractions of 2 Gy every other day), in two replicates [48]. After eliminating lncRNA hits with a TSS within 1 kb of any protein coding gene TSS, 466 lncRNAs were identified that decreased cell growth in the presence of radiation, and one lncRNA, *PVT1*, enhanced cell growth [48]. A “screen score”, as defined by the average phenotype and statistical significance of the top sgRNAs for each hit, was determined for lncRNAs negatively affecting cell growth in the radiation modifier screen as compared to the non-irradiated screen [48]. Thirty-three lncRNAs were prioritized based on their screen score as having significant sensitization effects in the presence of radiation [48].

Nine of these hits were further prioritized as lncRNA Glioma Radiation Sensitizers (lncGRS) based on their expression in adult GBM (U87, SF10360, SF10281) and pediatric diffuse intrinsic pontine glioma (DIPG) (SF8628, SF1021) cell lines, and ranked according to the ratio of screen scores between the radiation modifier screen and growth screen in non-irradiated cells [48]. The authors validated their findings through additional CRISPRi knockdown assays of a top candidate *lncGRS-1* in U87 cells [48]. A 42% decrease in cell proliferation upon CRISPRi knockdown of *lncGRS-1* was observed without radiation, and proliferation of cells expressing control sgRNAs decreased by 71% when exposed to radiation alone [48]. The authors report a synergistic effect upon CRISPRi knockdown of *lncGRS-1* in combination with radiation, which exhibited 95% decrease in proliferation, whereas a decrease of 83% would have been predicted by an additive effect model [48]. Further CRISPRi and ASO-mediated knockdown assays of *lncGRS-1* in patient-derived adult GBM SF10360 and pediatric DIPG SF8628 cultures exhibited reduced proliferation without radiation [48]. RNA-seq analysis of ASO-mediated *lncGRS-1* knockdown revealed glioma-specific activity in U87 and SF8628 cells, including up-regulation of genes involving p53, and down-regulation of genes controlling DNA damage and cell cycle functions [48]. ASO-mediated knockdown of *lncGRS-1* also sensitized glioma cells to radiation without reducing normal brain tissue viability as demonstrated ex vivo in mature brain organoids [48].

#### 2.2.3. MYCncLibrary Screen

Raffeiner et al. performed a CRISPRi screen to identify lncRNAs regulated by the MYC oncogene in Burkitt’s lymphoma (BL) [45]. The authors developed a CRISPRi library targeting 508 non-coding loci, including lncRNAs, over-expressed in accordance with MYC in the human lymphoid cell line P493-6 [45]. The library targeted 100 coding genes regulated by MYC and 14 genes coding for essential proteins as positive controls, as well as 100 non-targeting sgRNAs as negative controls [45]. Up to 10 sgRNAs were designed per gene, targeting within +/−400 bp around the TSS, for a total of 5708 sgRNAs [45]. Three replicates of this “MYCncLibrary” were screened in P493-6 cells and a BL cell line, RAMOS, expressing dCas9-KRAB as well as in the same cell lines not expressing dCas9-KRAB as negative controls [45]. Few sgRNAs were depleted upon comparison between the cell lines expressing and not expressing dCas9-KRAB, and although the greatest depletions were observed for the protein-coding control genes *EIF4A3* and *DNM2*, the authors questioned the effectiveness of the dCas9-KRAB system [45]. Upon testing different repressor domains coupled to dCas9, the most significant depletion of guides was found using SID [45]. These findings suggest that the optimal use of CRISPRi repressor domains may be cell type specific, and adds SID to the ever expanding dCas9 fusion protein arsenal [45].

As a result of the MYCncLibrary screen, a total of 109 ncRNAs were depleted in both cell lines, with 105 depleted in RAMOS only and 77 depleted in P493-6 only, suggesting their necessity in MYC driven cell proliferation [45]. Previously characterized lncRNAs including *MIR17HG*, *NEAT1*, and *DANCR* were identified as top hits driving P493-6 and RAMOS cell viability [45]. The top candidates *RAD51-AS1*, *ZNF433-AS1*, *TTN-AS1*, *SNHG17*, and *SNHG26* were further validated by CRISPRi fluorescently labeled growth assays [45]. These top hits included several antisense RNAs, which pose a limitation to the specificity of the screen results [45]. Raffeiner et al. acknowledge that the sgRNAs targeting *RAD51-AS1* also affect expression of the overlapping protein coding gene *RAD51*, which has previously been identified as necessary for cell viability [45]. However, the authors state that CRISPRi knockdown of *ZNF433-AS1* and *TTN-AS1* should not have an effect on the respective overlapping protein coding genes *ZNF433* and *TTN* based on RNA-seq data [45]. The small nucleolar RNA host genes *SNHG17* and *SNHG26*, chosen based on their TSSs at least (+/−) 10 kb away from other genes, were bound by MYC in promoter regions and are direct transcriptional targets of MYC [45]. Hits with strong sgRNA depletion over the course of the screen were also observed to have higher MYC binding, as detected by ChIP-seq [45].

CRISPRi has proven to be a revolutionary tool in functional lncRNA research. While a complete gene knockout is typically not achievable by CRISPRi, a significant knockdown is often sufficient to interrogate the function of the lncRNA of interest. A major limitation of currently available libraries is the genomic location of the targeted lncRNAs in respect to other genes. While it is feasible to exclude screen hits with adjacent protein-coding genes during the computational analysis, the possibility of deregulation of unintended targets remains.

### 2.3. CRISPRa

In addition to the CRISPR interference screens discussed above, GOF screens utilizing CRISPRa have been used to identify functional roles for lncRNAs in cancer [15,16]. CRISPRa allows for transcriptional activation of genes through the use of dCas9 associated with an activation domain, such as VP64 [35,44,49,50] (Figure 1D). The CRISPRa synergistic activation mediator (SAM) system consists of dCas9–VP64 coupled to a varied nuclear localization signal, synergistically functioning with the MS2 fusion protein fused to the transcription factor p65 and the activation domain human heat-shock factor 1 (NLS–dCas9–VP64 and MS2–p65–HSF1) [49]. SAM was optimally designed for a large scale screen of protein coding genes, but adequately upregulated lncRNA expression as well [49]. The optimal region for CRISPRa targeting is −1000 bp to −50 bp around the target TSS, with strongest gene activation between −400 to −50 bp [35,51]. Thus far, CRISPRa screens were utilized to identify lncRNAs involved in drug resistance [15,16].

#### 2.3.1. CRISPRa Screen for Vemurafenib Resistance Genes

The first large-scale CRISPRa screen targeting lncRNA loci in cancer identified resistance mediators to the BRAF inhibitor vemurafenib in the human melanoma cell line A375 [15]. The authors targeted 10,504 intergenic lncRNAs with CRISPRa-SAM, designing 10 sgRNAs per gene and 500 non-targeting controls for a total of 95,958 sgRNAs [10,15,52]. The target lncRNAs were identified by combining the RefSeq noncoding RNA catalog (filtered for transcripts longer than 200 bp and not overlapping RefSeq coding genes) with the Cabili lncRNA catalog, and filtered for lncRNA TSSs > 50 bp apart [10,15,52]. The sgRNAs were optimally designed within −800 bp to +1 bp around the target TSS [15]. A375 cells were transduced with the sgRNA library in four replicates and treated with vemurafenib alongside an untreated control for 14 days [15].

Comparison between vemurafenib treated and untreated cells revealed a significant difference in the distribution of sgRNAs targeting 16 lncRNA loci, of which five were reported to likely be false positives due to off-targeting effects of SAM [15]. The authors further validated six lncRNAs and explored the mechanisms by which they operate [15]. To investigate lncRNA function in trans, cDNA constructs encoding individual lncRNAs were over-expressed, resulting in no difference in drug resistance [15]. To identify lncRNA function in cis, the authors examined the expression of all nearby genes within 1 Mb of each lncRNA gene [15]. The authors conclude that the lncRNA *NR_109890* acts locally to up-regulate its neighboring gene, EBF transcription factor 1 (*EBF1*), which is involved in cell signaling, proliferation and survival [15,53]. However, the *NR_109890* TSS is within 1 kb of the *EBF1* promoter, indicating upregulation of *EBF1* may be due to off-target effects by SAM and not an up-regulation of *EBF1* by *NR_109890* [15].

#### 2.3.2. CRISPRa Screen for Cytarabine Resistance Genes

Bester et al. sought to identify genes involved in the resistance of acute myeloid leukemia (AML) to the chemotherapy treatment cytarabine (Ara-C) [16]. The authors utilized dual protein-coding and non-coding integrated SAM-mediated CRISPRa screens, targeting genes correlating with sensitivity or resistance to Ara-C in the AML cell line MOLM14 [16]. SAM-mediated activation was first validated in MOLM14 cells by targeting the promoters of known coding and non-coding genes, including the lncRNAs *MIAT* and *TUNA* [16]. The library targeted 14,701 lncRNA genes identified from the Human GENCODE V22 transcript annotations, merged with the Broad human lincRNA catalog, regardless of classification based on genomic positioning [10,16,35,49,54]. This non-coding library utilized four sgRNAs for each of the 22,253 lncRNA TSSs, for a total of 88,444 targeting guides and 99 non-targeting sgRNA controls [16].

Two replicates of the screen identified lncRNAs previously associated with cancer, including *TUG1*, *HOTAIRM1*, and *PVT1* [16]. Extensive analysis of the enriched targets and highly co-expressed protein-coding genes revealed genes involved in survival pathways known to affect leukemia and drug resistance [16]. Eleven significantly enriched and two depleted lncRNA candidates were validated through additional CRISPRa assays examining Ara-C response, as well as proliferation and survival assays in MOLM14 and HL-60 cells [16]. As expected, the enriched lncRNA targets provided the cell with Ara-C resistance and promoted increased survival, whereas depleted lncRNAs sensitized the cells to Ara-C [16]. The authors unveiled a novel mechanism by which a highly enriched gene pair from both screens, the lncRNA *GAS-AS2* and its associated protein coding gene *GAS6*, promote cellular survival in the presence of Ara-C by cis-regulation of the ligand coding GAS6 and trans-regulation of its receptor AXL [16].

The authors illustrate the genomic location of their chosen *GAS6/GAS-AS2* locus as divergent promoters within 100 bp [16]. CRISPRa is known to assert its effects within 1 kb of the TSS, thus further validation by means other than CRISPRa may have been preferable [35,49]. The authors confirmed their results by directing eight CRISPRa sgRNAs to *GAS6-AS2* and observed strong cellular resistance with two sgRNAs [16]. One of these directly targets *GAS6*, while the other sgRNA seems to have low on-target specificity, with potential binding sites on 15 different chromosomes. Experiments utilizing ASOs targeting *GAS6-AS2* in an Ara-C resistant leukemia cell line expressing *GAS6-AS2*, *GAS6*, and *AXL* show that knockdown of *GAS6-AS2* leads to decreased expression of the protein coding genes [16]. However, ASOs have been found to induce transcriptional termination in addition to post-transcriptional degradation [55]. This example highlights the importance of using more than one independent experimental method to confirm new functions of lncRNAs in cancer.

Transcriptional activation of an endogenous locus using CRISPRa is a unique method for identifying or validating functional lncRNAs in cancer within its genomic context. Identification of increased lncRNA expression conferring resistance to cancer drugs or overall cancer survival aid in the development of therapeutics. Similarly to CRISPRi, utilizing dCas9 removes concerns associated with DNA damage repair and associated toxicity. Improvements continue to evolve, making the activation system more efficient and CRISPRa applicable to functional GOF studies.

In summary, the advent of CRISPR/Cas technologies enabled functional screening to identify new therapeutic targets in cancer at unprecedented speed and precision. Hundreds to thousands of lncRNAs can now be screened in parallel using genomic knockout technologies via pgRNAs or splice site targeting approaches, or through transcriptional repression or activation of target genes under their endogenous promoters via CRISPRi or CRISPRa, respectively. One limitation of all CRISPR/Cas based approaches remains the incomplete and/or inaccurate genomic annotation of some lncRNAs; therefore, careful consideration is imperative when designing sgRNAs to TSSs. In addition, while the few studies carried out to date have mainly focused on cell growth and combinations with drugs or radiation, future readouts could include assays such as synthetic lethality, cell migration, invasion, metabolism, genomic stability or other attributes involved in the hallmarks of cancer.

## 3. Methods for Direct Targeting

While immense progress has been made using the new genome modification-based approaches, they are somewhat limited by potential effects on neighboring genes and the regulatory elements of the targeted DNA site, as well as the inability to effectively target lncRNAs that overlap other genes [56]. To navigate these caveats, direct targeting of the non-coding transcripts provides an alternative approach. Currently, there are three available methods for functional screening of lncRNAs through post-transcriptional targeting: RNA interference (RNAi), antisense oligonucleotides (ASOs) and CRISPR/Cas13 (Figure 2).

### 3.1. RNAi

RNAi was discovered in the 1990s and set the stage for current screening techniques [57,58]. The system utilizes a double-stranded RNA (dsRNA), which is cleaved into a single-stranded small interfering RNA (siRNA) by the RNase enzyme Dicer [59]. The siRNA then guides the multiprotein complex RNA-induced silencing complex (RISC) to the target RNA resulting in degradation by Argonaute 2 (Ago2) [57,58,60,61,62,63,64,65] (Figure 2C). Short hairpin RNAs (shRNAs) can also be utilized for RNAi, and are processed inside the cell to form siRNAs [66,67] (Figure 2C). Unlike siRNAs, shRNAs can integrate into genomic DNA, allowing for longer-term repression of target genes [66,67]. A number of high-throughput RNAi screens targeting lncRNAs for functional analysis in cancer have been performed over the past decade [68,69,70,71,72,73,74]. We will discuss the three most recent ones below.

#### 3.1.1. RNAi Screen Targeting Autophagy Related lncRNAs

In 2019, Tiessen et al. performed an siRNA screen targeting 638 lncRNAs (5 siRNAs per lncRNA target) in MCF-7 to identify lncRNAs functionally implicated in autophagy [68]. The siRNAs targeting *PLK1* and mTORC1-associated *RAPTOR* were used as positive controls, while siRNAs targeting *GFP* and the autophagy regulator *BECN1* served as negative controls [68]. Their MCF-7 cell line stably expressed GFP-LC3B, an autophagosome marker, which was used as a proxy for estimating autophagy levels through fluorescent microscopy quantification of GFP-LC3B puncta formation [68]. The authors used the total cell count as a metric of cell viability [68]. Following transfection with the siRNA library, cells were incubated for 72 h prior to fixation for fluorescence microscopy analysis [68]. SiRNAs that negatively affected cell viability were excluded for downstream analysis [68]. The authors identified 63 lncRNA transcripts that significantly affected the number of GFP-LC3B puncta, suggesting that they played a regulatory role in autophagy [68]. Validations were performed on the top 14 hits using the two best scoring siRNAs per gene from the initial screen [68]. The top candidate, downregulated RNA in cancer (*DRAIC*), was selected for further functional analysis and characterized as a novel regulator of autophagy [68].

#### 3.1.2. RNAi Screen Targeting lncRNAs Important for Cell Division

Most recently, Stojic et al. performed an RNAi screen targeting 2231 lncRNAs in HeLa cells (4 siRNAs per lncRNA) to identify lncRNAs impacting cell division [69]. Negative control siRNAs, siRNAs targeting the protein-coding gene *Ch-TOG/CKAP5*, and cells without any siRNA treatment were used as controls [69]. Two RNAi screens were performed in parallel (Screens A and B) using antibodies to label microtubule cytoskeleton, centrosomes and nuclei [69]. Screen A additionally labelled the actin cytoskeleton and Screen B labelled mitotic cells [69]. The cells were fixed after 48 h and processed for immunostaining and image acquisition [69]. Automated image analysis was used to segment the cells followed by quantification of mitotic progression, chromosome segregation and cytokinesis using an in-house developed pipeline [69]. Subsequent validation targeting their top 25 lncRNA candidates confirmed their findings [69]. The authors chose one candidate, *linc0089*, for detailed molecular studies, including phenotype validations using ASOs, CRISPRa and rescue assays, revealing *linc0089* as a key player in controlling mitotic progression [69].

#### 3.1.3. RNAi Screen In Vivo

RNAi screens have been widely performed in vivo to identify oncogenic protein-coding genes [75,76,77,78,79,80,81,82,83,84,85]. However, only one large-scale in vivo screen has been performed for lncRNAs to date [84]. Delás et al. introduced an RNAi library targeting 120 lncRNAs (>4 shRNAs per lncRNA) into a mouse model of AML [84]. ShRNAs targeting *Renilla* luciferase and Replication Protein A3 (*Rpa3*) were used as negative and positive controls, respectively [84]. MLL-AF9/NRAS^G12D^ AML cells were infected with the shRNA library in vitro, followed by tail-vein injections [84]. When analyzing shRNA representation from pre-injection pools and bone marrow samples taken at the endpoint of the study after 14 days, the authors identified 20 hits that were significantly depleted [84]. To validate their results, the authors performed competitive proliferation assays in vitro for all candidates, with 14/20 lncRNAs showing a depletion of >50% [84]. Similar proliferation assays were performed on nine of these lncRNAs in the murine breast cancer cell line 4T1 to test for cell type specificity of the candidates [84]. None of the nine lncRNAs were depleted in 4T1 cells, indicating once again that lncRNAs act in a highly cell type specific manner [84].

RNAi, specifically using shRNAs, is a relatively accessible, inexpensive and easy to scale method. Its widespread use over the years demonstrates the potential to successfully identify new drivers of cancer progression. The most recent screens discussed above further highlight more advanced methods of assay readout, such as in vivo applications and the identification of lncRNAs impacting autophagy and cell division, compared to CRISPR/Cas9 screens that are still in their infancy. However, RNAi does come with limitations such as the pervasive off-target effects of RNAi [86,87,88,89,90,91,92]. It has been shown that siRNAs can function as microRNAs (miRNAs) in the miRNA-mediated translation repression pathway, resulting in non-specific knockdown of dozens to hundreds of transcripts [93]. While all current functional screening methods carry the risk of off-target activity, studies comparing CRISPR technologies to RNAi have found the off-target effects to be more substantial in RNAi [88,89]. Another pitfall of RNAi as a functional analysis tool for lncRNAs is that RNAi activity is limited mainly to the cytoplasm, which makes targeting nuclear lncRNAs difficult [94].

### 3.2. ASOs

ASOs are short (12–20 nt) synthetic single-stranded oligonucleotides that bind to complementary RNA sequences with high specificity [95]. When bound to their target RNA, ASOs induce RNase-H mediated degradation resulting in gene knockdown [95] (Figure 2A). ASOs bind to their target molecule post-transcriptionally, suggesting that any observed phenotype can be attributed to the targeted RNA. However, recent evidence shows ASOs acting co-transcriptionally and triggering premature transcription termination [55]. This novel finding implies that the potential effects of ASOs on transcription should be considered when interpreting functional studies [55]. ASOs are preferable over RNAi for post-transcriptional targeting of lncRNAs because they efficiently reduce both nuclear and cytoplasmic targets [96,97,98,99].

While ASOs have been used to perform smaller scale lncRNA screens previously [100], the first ASO screen targeting hundreds of lncRNAs was published recently [101]. Ramilowski et al. included 2021 ASOs targeting 285 lncRNAs (>5 ASO per lncRNA) in non-transformed primary human dermal fibroblasts, obtaining a median knockdown efficiency of 45.4% across all screened ASOs [101]. A positive control ASO targeting *MALAT1* and a negative control ASO (NC_A, Exiqon) were used [101]. The authors were able to target 194 of the 285 lncRNAs with at least two ASOs [101]. ASO transfected cells were imaged every 3 h for a total of 48 h to assess changes in cell growth, where 15 out of the 194 lncRNAs exhibited an altered growth phenotype [101]. In addition to characterizing cell viability, they measured changes in cell morphology using a machine learning-assisted imaging workflow in which each cell was segmented and its individual morphological features quantified [101]. The authors suggest that assessing the phenotype at a molecular level is an important asset, as some lncRNAs may not be essential for cell viability but alter the observed phenotype in other ways [101]. For instance, 14 of the 194 lncRNAs impacted the spindle-like morphology of fibroblasts without any reduction in cell growth [101]. Overall, 59 of the 194 lncRNA targets affected cell growth and/or morphology [101].

ASO screens are a valuable asset to the field but can be more costly and labor intensive compared to other approaches of functional screening. In general, ASOs are more commonly applied for validation of targets rather than primary screens. Similar to siRNAs, ASO-mediated lncRNA knockdown is short lived and not suitable for long-term viability screens. For instance, the duration of Ramilowksi et al.’s ASO screen was only 48 h compared to most CRISPR viability screens that are carried out over 20–30 days. Additionally, as with all screening approaches, it is advisable to use multiple independent ASOs targeting the same gene to validate the observed phenotype (as reviewed in [102]). A minimum of two independent ASOs per gene is advised to account for any potential off-target effects (as reviewed in [102]). An advantage of ASO screens over CRISPR screening techniques is their potential for direct translation to the clinic. While gene therapy using CRISPR may be a viable therapeutic option in the future, several ASO drugs are already FDA approved and many more are in clinical development [103,104,105]. Preclinical studies show ASOs as promising therapeutics for lncRNAs in cancer [37,100,106].

### 3.3. CRISPR/Cas13

CRISPR/Cas13 is the most recently developed technique to interrogate lncRNA functionality post-transcriptionally [107,108]. Cas13, an endoribonuclease also known as C2c2, operates to cleave target single-stranded RNA (ssRNA) as guided by a sgRNA [107,109,110] (Figure 2B). CRISPR/Cas13 manipulation targets RNAs with low off-target effects and controllable specificity of target knockdown [107]. Target binding can be abolished by as few as 1–2 nucleotide mismatches in the center of the sgRNA, with varying degrees of relative target knockdown due to mismatches in every position but the 5′ and 3′ ends [107]. Importantly for lncRNA research, CRISPR/Cas13 has been shown to successfully knock down transcripts located within the cytoplasm and nucleus [107,108].

To demonstrate the feasibility of CRISPR/Cas13 functional lncRNA screening, Xu et al. targeted very long intergenic non-coding RNAs (vlincRNAs) [108]. The authors used doxycycline inducible Cas13 to screen 25 vlincRNAs upregulated in the presence of 3 drugs (etoposide, mirin, and imatinib) in K562 cells, along with 10 mRNAs biologically relevant to CML [108]. For each target, sgRNA pairs were designed consisting of one complementary guide and a corresponding non-targeting control with 3 centrally located bp mismatches, for a total of 10 sgRNA pairs per vlincRNA [108]. Cells stably expressing Cas13 were infected with the sgRNA library at an infection rate of 24% and treated with the drugs mentioned above to select for resistance [108]. Drug treatment and recovery were performed in the presence and absence of doxycycline, allowing for a control screen in the absence of active Cas13 [108]. Four layers of controls and stringent analysis suggest that the phenotypic readout of this study is indeed due to the functional nature of the targeted lncRNAs [108]. The authors found 64% of the targeted vlincRNAs to be significantly depleted or enhanced, thereby functionally relevant in CML drug resistance [108]. The large percentage of functional vlincRNAs identified in this survival screen is likely due to the initial selection of targets under the same applied stress conditions [108].

While this initial study was comparably low-throughput, it serves as a proof-of-concept study for Cas13 as a functional screening platform for lncRNAs in cancer [108]. One advantage of using CRISPR/Cas13 over other post-transcriptional knockdown strategies such as RNAi or ASOs is the ability of stable long-term expression of both Cas13 and sgRNAs, allowing for longer time-course studies [107]. As opposed to genomic Cas9 approaches, the results of reverse-genetic engineering by Cas13 can be directly attributed to the functionality of the targeted transcripts [107,108,111]. Overall, the CRISPR/Cas13 approach is a promising recent addition to functional screening methods in cancer.

In summary, currently available methods for direct lncRNA targeting include RNAi, ASOs and CRISPR/Cas13. When considering the different approaches, it is important to note that both CRISPR/Cas13 and ASOs are effective in the cytoplasm and nucleus, while RNAi can generally only be utilized in the cytoplasm. Furthermore, siRNAs and ASOs are suitable for short term studies only, while shRNAs and CRISPR/Cas13 enable long-term studies similar to CRISPR/Cas9 genomic and transcriptional targeting approaches. In general, post-transcriptional targeting is a more direct approach than genomic or transcriptional based screening, can be applied to all lncRNAs regardless of their genomic position or intron-exon structure, and the results are less likely to be impacted by unwanted effects on adjacent genes. In addition, identified potent siRNAs and ASOs have potential for quick translation to preclinical and clinical development. Moreover, previous studies indicate that RNAi has higher off-target rates compared to CRISPR-based systems, potentially leading to incorrect conclusions. While direct comparisons of ASO off-targets are still lacking, ASO based functional screens are also challenging to scale up and expensive.

## 4. Future Directions

Recent years have seen important progress in developing screening methods to identify functional lncRNAs in cancer. However, the vast majority of approaches are currently limited to 2D cell culture. While a convenient starting point for large-scale screens, immortalized cell lines grown in 2D do not fully represent the heterogeneity and pathophysiology of patient tumors (as reviewed in [112,113]). Due to the cell type- and contextual specificity of lncRNA function, research models to identify lncRNA drivers of cancer growth should closely reflect patient tumors and the interplay of systems in which the tumor resides. Recent technological advancements enable genetic manipulation and functional screens in in vitro 3D spheroids, ex vivo organoids derived from primary tissue or stem cells, and in vivo models that portray malignant tumors more accurately [114,115,116,117]. We will discuss some of the most recent advances and their potential future applications for lncRNA studies below.

### 4.1. Spheroids

In vitro 3D spheroids are immortalized cell lines grown suspended in an artificial extracellular matrix such as collagen or Matrigel [114]. Han et al. performed a genome-wide CRISPR-Cas9 screen targeting ~21,000 protein coding genes in the lung adenocarcinoma cell line H23, grown both as 3D spheroids and in 2D monolayers [114]. Comparison of the screen in 2D and 3D revealed divergent phenotypes between the two models, with hits identified in spheroids representing patient tumors more closely [114]. The authors further validated the top hits in 2D, 3D, and subcutaneous xenograft mouse tumors, which further confirmed the superiority of spheroids and xenografts compared to 2D culture [114]. This study highlights the importance of considering 3D cultures in cancer biology and suggests that functional screens in spheroids may be more closely mimicking the tumor transcriptome than 2D cultures.

### 4.2. Organoids

Ex vivo organoids are 3D models derived from primary tissue or tissue stem cells, and artificially enticed by extensive culturing methods to differentiate into organ-like structures [116,117,118]. Mature organoids are composed of multiple cell types that exhibit self-renewal and self-organization properties, and can be co-cultured with immune cells to mimic parts of the microenvironment [119]. Recently performed large-scale CRISPR/Cas9 knockout screens targeting protein-coding genes uncovered a number of technical issues associated with genome-wide screens in human organoids [116,117]. One major challenge is the feasibility of maintaining the appropriate cell number required for sufficient library coverage [117]. The intrinsic cell heterogeneity of organoids provides a further obstacle to successful genomic modification [116]. Another caveat of CRISPR/Cas9 screening in organoids is the variability of phenotypic strength and penetrance of sgRNAs. Michels et al. compared the activity of sgRNAs side by side in HepG2 cells and human colon organoids, revealing that only a fraction of sgRNAs were active in organoids [116]. This suggests that currently available sgRNA design algorithms are optimized for 2D cell culture, and improvements to the software are required for successful CRISPR/Cas9 screening in organoids [116]. Furthermore, the same study showed that pooled screening in organoids can be prone to false positives due to strong expansion of outlier clones [116].

### 4.3. Animal Models

To identify new lncRNA drivers of cancer growth, the most direct and likely best practice approach would be a functional screen in vivo in animal models, as they have the potential to overcome technical constraints of 2D and 3D cultures. They contain a functional tumor microenvironment and retain the ability to develop metastasis to distant organs, enabling the identification of drivers of invasion and metastasis in addition to drivers of primary tumor growth [76,82,84,120]. However, the feasibility of in vivo screens is limited by restricted target tissue accessibility, library size and coverage, as well as high associated costs. An in depth review of these limitations can be found in [121]. In the context of investigating lncRNAs, lack of sequence conservation of the intended targets is another important consideration. One approach for circumventing these issues is in vivo screening in xenografted human cells or organoids [76,82,84,116,120]. Transplantation techniques come with their own set of constraints, such as requiring immunodeficient mice and restrictions on how many cells can be transplanted, limiting either library size or coverage (as reviewed in [121]). Alternative approaches include the use of humanized mouse models. In 2020, Ruan et al. developed a pipeline to functionally analyze non-conserved lncRNAs in vivo using a liver-specific humanized mouse model [115,122]. While the system may require further optimization, Ruan et al. were able to identify and characterize the role of *LINC01018* in liver metabolism using shRNA knockdown in vivo [115].

Although limitations remain and future studies are required for further optimization, the functional screens described here pave the way for high-throughput screening in spheroids, organoids and in vivo. While no large-scale CRISPR/Cas9 screens targeting lncRNAs have been performed in these models, they have the potential to be utilized to successfully identify clinically relevant oncogenic lncRNAs in the future. These models also open the door for additional future investigations of phenotypic readouts as a result of lncRNA function in characteristics of cancer that may be difficult to uncover in 2D models. Future studies will unveil lncRNAs driving additional hallmarks of cancer, such as cell invasion or migration, in models most closely indicative of patient cancer to further contribute to the advancement of cancer treatments.

## 5. Conclusions

High-throughput functional screens are an invaluable asset to the field of cancer genomics. Previous identification of lncRNAs for therapeutic targeting centered around differential expression analysis, however, candidate selection on this basis does not imply function. Functional screens supersede differential expression studies when identifying lncRNAs involved in cancer as seen by contrasting disruption in proliferation, regardless of lncRNA expression, between different cancer type models. Current screening approaches allow for the ability to assess the function of thousands of lncRNAs in parallel and have solidified the concept of lncRNAs as therapeutic targets in cancer. The increasing repertoire of screening methods allows for the option to perform orthogonal approaches for functional analysis. As there are advantages and disadvantages of each technique, using multiple approaches to determine the function of a lncRNA allows for correction against false-positives or false-negatives specific to a single method.

There are some common caveats and considerations of the mentioned screening methods. Firstly, all of the described methods rely on base complementarity with their target lncRNA sequence. This creates the obvious complication of potential off-target effects due to sequence similarities between genes. Careful consideration is required when designing and determining an appropriate number of oligonucleotides (gRNAs, siRNA/shRNAs and ASOs) to be screened for each target, to account for experimental error. Secondly, all functional screens should be performed in multiple relevant cell lines in order to identify pertinent lncRNA targets of a particular cancer type, as opposed to identifying lncRNAs functioning in one specific cell line. Several of the discussed studies showed independently that lncRNA function is highly cell type- and context dependent. These caveats must be considered but should not detract from the overall benefits that these screening techniques have on the advancement of lncRNA functional analysis in cancer research. Functional screens of lncRNAs in cancer will continue to identify numerous additional therapeutic targets, as lncRNAs have been shown to drive important mechanisms in cancer, such as proliferation and invasion. ASOs or RNAi approaches are viable approaches to target lncRNAs clinically in the future.

## Figures and Tables

**Figure 1 cancers-12-03695-f001:**
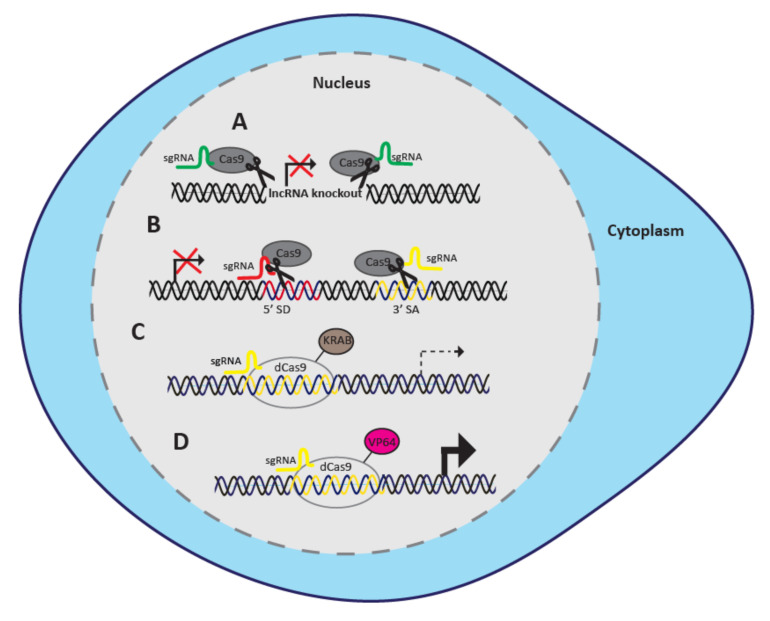
Methods for genomic and transcriptional targeting. (**A**) Paired-guide RNAs recruit Cas9 to the target lncRNA gene resulting in a complete knockout. (**B**) SgRNAs direct Cas9 to the splice sites of lncRNA genes resulting in complete knockout. (**C**) A sgRNA tethers dCas9-KRAB to the TSS resulting in gene knockdown. (**D**) A sgRNA guides dCas9-VP64 to the TSS resulting in transcriptional activation.

**Figure 2 cancers-12-03695-f002:**
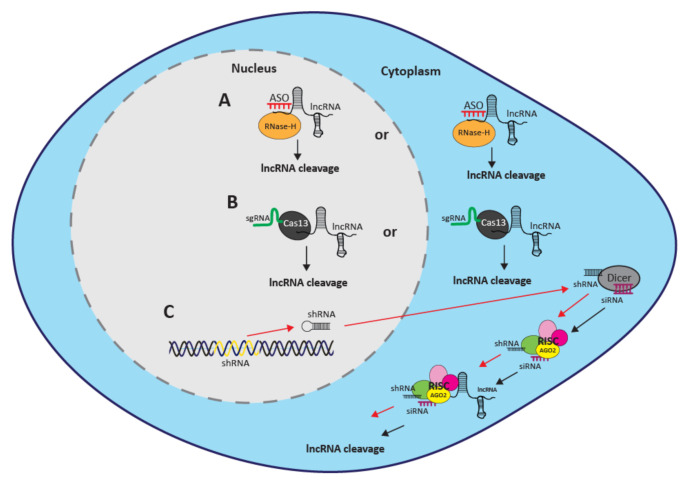
Methods for direct targeting. (**A**) Binding of the ASO to the lncRNA transcript leads to RNase-H mediated degradation resulting in post-transcriptional knockdown, which can occur in either the nucleus or the cytoplasm. (**B**) A sgRNA guides Cas13 to the lncRNA transcript resulting in post-transcriptional knockdown, which can occur in either the nucleus or the cytoplasm. (**C**) Red arrows depict shRNA mediated RNAi in which shRNAs are integrated into the genomic DNA prior to transport to the cytoplasm. Black arrows depict siRNA mediated RNAi in the cytoplasm. RNase Dicer cleaves double stranded RNA to generate single-stranded RNAs. The single-stranded RNAs form a complex with RISC and the lncRNA, resulting in degradation of the transcript by Ago2.

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
