# Peer review of "Functional Screening Techniques to Identify Long Non-Coding RNAs as Therapeutic Targets in Cancer"

_cancers, 2020, doi:10.3390/cancers12123695_

Round 1
Reviewer 1 Report
The manuscript entitled "Functional screening techniques to identify long non-coding RNAs as therapeutic targets in cancer" presents a timely and comprehensive review of the literature on an important topic. Prior to publication, I have two suggestions:
1) The authors mention a number of studies in the text which are based e.g. on lncRNA siRNA screens, but which are not listed in table 1. To give a short but comprehensive perspective also in this table, all studies should be listed there, as well.
2) In the conclusions, the authors may want to add one paragraph about the development of lncRNAs as therapeutic targets based on functional screens since this was also mentioned in the abstract.
Author Response
We would like to thank the Reviewer for their very useful comments.
1) We have amended Table 1 with the siRNA studies as requested.
2) We have added two sentences at the end of the conclusion regarding lncRNAs as therapeutic targets in cancer (lines 649-652).
Reviewer 2 Report
Lucere and colleagues present a useful review on the rapidly developing field of screening for lncRNA hits in cancer. The authors present a wide and deep summary of the field that will be an important guide for many researchers likely to enter this area in the coming years. My comments are relatively superficial.
Line 30: On the topic of functional cancer mutations in lncRNAs, it would be most appropriate to cite the Lanzos et al paper on the subject PMID 28128360.
L53: "seek targets that are decreased," this is rather unclear. If we are discussing more common LOF screen: In a dropout screen, it is the LOF constructs targething those genes that are decreased (rather than the target genes themselves) - the genes themselves promote the phenotype in question. Similarly the explanation in the next line for positive screens is unclear. Positive screens search for LOF constructs that increase over time, hence targeting genes that repress the phenotype. Recommend the authors rewrite for clarity.
L79: "which introduces premature stop codons" - or simply create nonsense protein.
L433: "RNAi is a relatively inexpensive and easy to scale method" This is debatable. siRNA screens are expensive to create or buy, and scaling requires robotic expertise beyond many labs. shRNA on the other hand is more accessible.
Regarding CRISPR deletion: The authors omit some important points. Firstly, by introducing pairs of double strand breaks (highly toxic to cells) this approach runs the risk of non-specific selection effects, particularly for regions of high copy number. For example, a triploid region targeted in this way would have potentially 3 x2 = 6 DSBs. Therefore, validation by alternative methods and filtering out high copy regions is strongly advised. Also, the toxicity of DSBs means that it is preferable for negative control guides to be targeting (somewhere in the genome) rather than non-targeting (completely random). Second, CRISPR deletion can have unknown numbers of false positives by deleting an important DNA region. This makes validation by other methods that directly perturb RNA very important.
ASOs: A few important things that were not mentioned. First it is strongly advised to replicate results with at least 2 independent ASOs to ensure on target effect - see Corey paper PMID 30907681. Second, ASOs are considered gold standard for validating screens, rather than for performing primary screen. If you see a lncRNA phenotype with ASO, you can be pretty sure that the effect is mediated by either mature RNA or else the process of producing it.
Zhu et al (Wensheng Wei lab) deletion screens: These screens have been reinterpreted in a couple of follow up papers and criticised on a number of counts including their sensitivity - perhaps worth mentioning: Bergada et al PMID 31681950 and Horlbeck et al PMID 32094656
LncRNA annotation and screening: Both CRISPR deletion and CRISPRi act on a narrow distance window. LncRNA annotations are notoriously poor in terms of missing genes, and the accuracy of 5' annotation. This strongly impacts the ability to achieve LOF. Recommend to discuss this and contrast with protein coding genes that are far more straightforward.
Author Response
We would like to thank the Reviewer for their thoughtful and detailed comments, which greatly contribute to the revised version of our Review.
1) We added the Lanzos et al paper in line 34.
2) We agree that the section was not as clear as it should have been, and re-wrote it (lines 56-61).
3) While we agree that our statement could be simplified, we prefer our more detailed explanation for additional clarity. We hope this will be useful for researchers not yet familiar with the field.
4) We agree, and have amended accordingly in line 466.
5) The reviewer is absolutely right, and we have added information with regards to high copy number regions in lines 170-183. In addition, we further emphasized the need to validate CRISPR based approaches using direct transcript targeting in this new paragraph. We further amended lines 163-165, discussing targeting vs non-targeting negative control gRNAs.
6) Yes, absolutely. We highlighted that two or more ASOs have to be used, and added that ASOs are mostly used in validation rather than screening approaches in lines 505-512.
7) We would like to thank the Reviewer for bringing these important discussions to our attention. We added comments regarding these in lines 128-132, and cited the suggested papers.
8) Excellent point. We added the challenges around lncRNA annotations as discussion points in lines 93-96, and 391-393.